# Maya Blue Used in Wall Paintings in Mexican Colonial Convents of the XVI Century

**Luisa Straulino-Mainou** [1,*], **Teresa Pi-Puig** [2,3,*], **Becket Lailson-Tinoco** [4], **Karla Castro-Chong** [4], **María Fernanda Urbina-Lemus** [5], **Pablo Escalante-Gonzalbo** [6], **Sergey Sedov** [2] and **Aban Flores-Morán** [7]

[1]  Coordinación Nacional de Conservación del Patrimonio Cultural, Instituto Nacional de Antropología e Historia (INAH), General Anaya S/N Ex, Av. del Convento, San Diego Churubusco, Coyoacan, Mexico City 04120, Mexico

[2]  Instituto de Geología, Universidad Nacional Autónoma de México (UNAM), Cd. Universitaria, Coyoacán, Mexico City 04510, Mexico; sergey@geologia.unam.mx

[3]  Laboratorio Nacional de Geoquímica y Mineralogía (LANGEM), Universidad Nacional Autónoma de México (UNAM), Cd. Universitaria, Coyoacán, Mexico City 04510, Mexico

[4]  Facultad de Ciencias Sociales y Humanidades, Universidad Autónoma de San Luis Potosí, Alvaro Obregon 64, Centro, San Luis, S.L.P. 78300, Mexico; becket.lailson@uaslp.mx (B.L.-T.); karla.castro@uaslp.mx (K.C.-C.)

[5]  Escuela Nacional de Conservación Restauración y Museografía, San Diego Churubusco, Coyoacan, Mexico City 04120, Mexico; fernanda_urbina_l@encrym.edu.mx

[6]  Instituto de Investigaciones Estéticas, Cd. Universitaria, Coyoacán, Mexico City 04510, Mexico; pabloeg@unam.mx

[7]  Centro de Enseñanza Para Extranjeros, Cd. Universitaria, Coyoacán, Mexico City 04510, Mexico; aflores@cepe.unam.mx

*   Correspondence: luisa.straulino@inah.gob.mx (L.S.-M.); tpuig@geologia.unam.mx (T.P.-P.); Tel.: +52-55-5622-4283 (ext. 207) (T.P.-P.)

**Abstract:** Maya blue is a well-known pre-Hispanic pigment, composed of palygorskite or sepiolite and indigo blue, which was used by various Mesoamerican cultures for centuries. There has been limited research about its continued use during the Viceroyalty period; therefore, the sixteenth century is the perfect period through which to study the continuity of pre-Hispanic traditions. The fact that the indigenous people were active participants in the construction and decoration of convents makes their wall paintings a good sampling material. X-ray fluorescence (XRF), scanning electron microscopy (SEM) and X-ray diffraction (XRD) were performed in samples of blue found in convents across Puebla, Tlaxcala and Morelos in order to identify whether the numerous hues of blue were achieved with Maya blue or with other pigments. We found no copper (Cu) or cobalt (Co) with the XRF, so several pigments, such as azurite, smalt or verdigris, were discarded. With SEM, we discovered that the micromorphology of certain blues was clearly needle-shaped, suggesting the presence of palygorskite or sepiolite. In addition, we found silicon (Si), magnesium (Mg) and aluminum (Al) by using energy-dispersive X-ray spectroscopy (EDS) in all blue samples, which also suggests the presence of these magnesium-rich clay minerals. With the XRD samples, we verified that the blues were produced with these two clay minerals, thus confirming that several wall paintings were manufactured with Maya blue. These findings confirm that this particular manmade pre-Hispanic pigment, Maya blue, was an important pigment prior to the Viceroyal period.

**Keywords:** Maya blue; wall paintings; sixteenth century; palygorskite; X-ray fluorescence; X-ray diffraction

## 1. Introduction

The arrival of Spaniards in the Mesoamerican territory meant an interaction of two cultures that were unknown to each other. This contact made societies converge in all aspects. Both cultures' presence can be distinguished throughout the XVI century to a

greater or lesser extent within institutions, beliefs, buildings and objects. Among these elements, the case of convents stands out since the interaction of cultures took place (more intensely) there, and the creativity in which it was expressed exceeded the limits of what was imagined. The reasons for this convergence were clear: first, in this place, the integration of the indigenous culture into the western was carried out in a planned way; second, the indigenous people, being the ones who constructed these buildings, portrayed their cosmovision in them. The clearest example of this is in the wall painting decoration since jaguars, eagles, native plants and pre-Hispanic symbols interact with the Christian saints and scenes of Jesus' life. This interaction also occurred in the techniques used for this type of decoration, since the indigenous people who were painting the walls introduced their techniques and materials. Thus, we can see in the murals how the oxide red, the black smoke, the lepidocrocite and the Mayan blue produced an Indigenous–Christian image [1–6] (Figure 1).

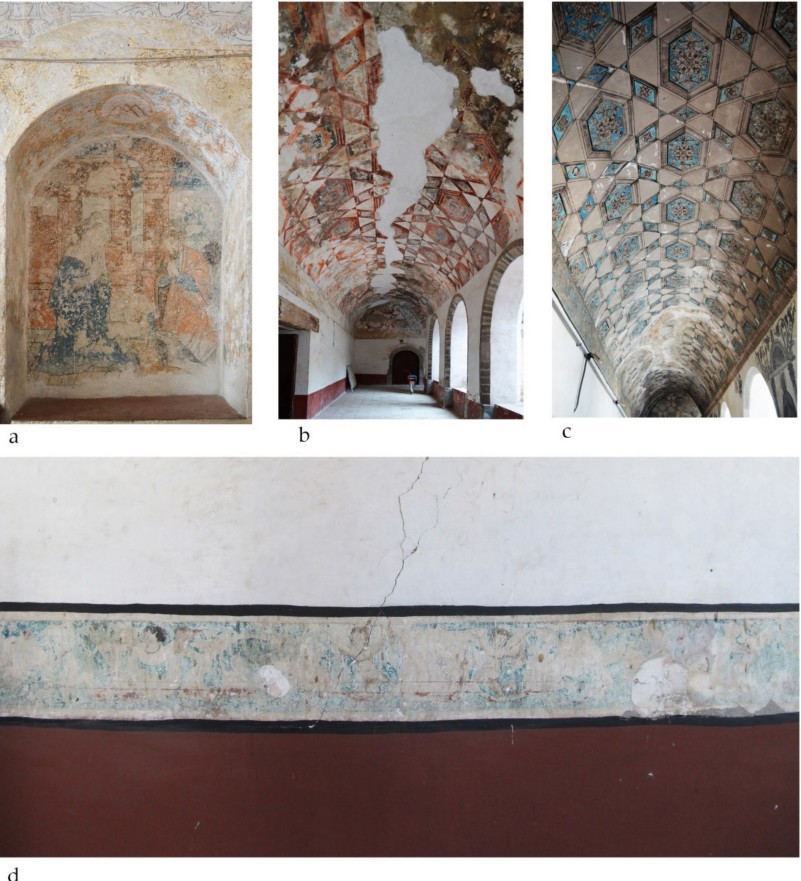

**Figure 1.** Examples of different hues of blues used in XVI century wall painting. From these wall paintings, very small samples were taken for XRF, SEM-EDS and XRD: (**a**) Tezontepec, (**b**) Ocuituco, (**c**) Oaxtepec, lower cloister and (**d**) Oaxtepec, upper cloister.

Among these colors, Maya blue is a special case. Maya blue is a synthetic pigment made from a fibrous clay mineral (palygorskite, with international mineralogical association (IMA) approved formula $(Mg,Al)_2Si_4O_{10}(OH)\cdot4H_2O$ [7]), plus an organic colorant (indigo: $C_{16}H_{10}N_2O_2$) [8]. There is crystallographic evidence that the Maya blue from Central Mexico used a different palygorskite compared to the Yucatan one. This is because Yucatecan palygorskite is a mixture of almost equal parts of monoclinic and orthorhombic palygorskite, while the Maya blue of Central Mexico exhibits mostly the orthorhombic form. However, some studies conducted in Central Mexico have identified a different Maya blue pigment made with sepiolite instead of palygorskite in archaeological objects [9].

Maya blue was used from the fifth century AD to colonial times in Mexico; nevertheless, there have been some studies that placed Maya blue in Cuba during its colonial period, identified as the pigment in a color formerly known as "Havana blue". This blue pigment was used mostly in decorations dated from the middle of the eighteenth century to around 1860 [10,11].

There has been some evidence of the use of Maya blue in XVI century codices such as *La Historia General de las Cosas de la Nueva España* by Fray Bernardino de Sahagún [12], in some maps (Ameca, Atlatlauca, Ixtapalapa, Mextitlán and Tehuantepec) of the *Relaciones Geográficas* made between 1578 and 1583 [13] and in the Cuauhtinchan 2 Codex, in which the indigo colorant was detected with Raman spectroscopy and the palygoskite with Fourier transform infrared spectroscopy (FTIR) [14], as some examples. Additionally, some cases of Maya blue used in XVI century convents have been identified in Jiutepec (Morelos), Actopan (Hidalgo), Epazoyucan (Hidalgo), Cuahutinchan (Puebla), Tezontepec (Hidalgo) and Totimehuacán (Puebla) [8,9,15].

Furthermore, some non-invasive studies, such as those using UV lighting, surface-enhanced Raman spectroscopy (SERS), X-ray fluorescence and FTIR, have been conducted on XVI century convents of the Augustinian order. With these techniques, indigo was found in the wall paintings of Actopan, Epazoyucan and Ixmiquilpan; nevertheless, Wong (2014) reported that the data acquired with FTIR were not of sufficient quality and thus they were not used; in consequence, it then was not possible to identify palygorskite, and therefore Maya blue could be the pigment used for the blues that exhibited indigo [16,17].

The search for Maya blue is further complicated by the fact that a number of other blue pigments were used in colonial times. Some of these pigments were introduced by the Spanish people and others were known in pre-Hispanic times. Among them are ultramarine (lazurite mineral), azurite (copper carbonate) and smalt blue (glass with cobalt). Some other blues were incorporated later on with the development of chemistry in the XVIII and XIX centuries, such as Prussian blue (ferrous ferrocyanide), cobalt blue (cobalt aluminate), cerulean blue (cobalt stannate), Bremen blue (copper hydroxide with copper carbonate) and manganese blue (manganate crystals mixed with barium sulfate) in the XX century [18–20]. These pigments could not only substitute but also occur together with the Maya blue due to repainting or restorations.

The aim of this paper is to identify the Maya blue pigment in the blue hues on several XVI century wall paintings in Central Mexico and to discriminate it from other pigments with similar tonalities. We propose to use palygorskite and/or sepiolite as an indicator of Maya blue's presence, whereas high concentrations of some specific elements (Cu, Co, Fe, etc.) could evidence the use of different pigments. To obtain these indicators, we applied the following methods: portable X-ray fluorescence equipment, scanning electron microscopy with energy-dispersive spectroscopy and X-ray diffraction. All of these techniques are well established to characterize pigments used in wall paintings [21–24]. The presence of these clay minerals in blues, in combination with other characteristics of the samples (such as absence Cu or Co minerals), is an almost certain indication of the used pigment being Maya blue. Moreover, we acquired information about the pictorial technique in which the Maya blue was applied.

## 2. Materials and Methods

The study was focused on the Central Highlands of the current Mexican territory, on the convents founded by the mendicant orders in the states of Hidalgo, Tlaxcala, Morelos, Puebla, Estado de México and Mexico City. This territorial delimitation is due to the fact that, particularly in these locations, the mendicant orders (especially the Franciscans and Augustinians) sought to integrate indigenous cultures into the new developing society. On the other hand, the region was inhabited by groups affiliated with the Nahua and Otomí, which had a particular way of creating a unique image and cosmovision, thus producing art that was different from that seen in Oaxaca, Michoacán or the Mayan area. With this delimitation, we restricted the study to 67 foundations which had wall painting

remains—from small fragments to complete programs (Appendix A shows all convents with wall paintings in the Central Highlands of Mexico). Due to time constraints, only 35 were broadly studied with XRF, and for conservation purposes, small samples were taken only in 10 of them.

### 2.1. X-ray Fluorescence (XRF)

The palette colors of 35 convents were studied with the portable XRF equipment, collecting preliminary data before taking physical samples to define the best locations for sampling. In these 35 convents, we observed a palette of 13 colors with different hues plus the color given by the plasters and stuccos. Twenty-eight of the convents had blue colors, and 2 had greenish blues, possibly composed of Maya blue (Table 1).

**Table 1.** List of convents whose paintings were studied by XRF, their location, foundation year and color of pigments sampled.

| Convent | State | Year | st | ye | or | bl | br | cr | fl | gr | pr | bl | rd | bg | pk | gr | pl |
|---|---|---|---|---|---|---|---|---|---|---|---|---|---|---|---|---|---|
| 1. Yecapixtla | Morelos | 1535 | x | x | – | x | x | – | – | – | – | x | x | x | – | –x | x |
| 2. Atlatlahucan | Morelos | 1569 | x | – | x | – | – | – | – | – | – | x | – | x | – | x * | x |
| 3. Izúcar de Matamoros | Puebla | 1540 | x | – | x | – | x | – | x | – | – | x | x | x | – | x | – |
| 4. Atlixco | Puebla | 1550 (c.) | x | – | x | x | – | – | – | – | – | x | x | – | – | – | – |
| 5. Ixmiquilpan | Hidalgo | 1550 | x | x | x | x | – | – | x | – | – | x | x | – | – | x | x |
| 6. Tochimilco | Puebla | 1569 | – | – | x | – | – | – | x | – | – | x | x | – | – | x | – |
| 7. Cuernavaca | Morelos | 1552 | x | – | x | x | x | – | – | – | – | x | x | – | – | x | – |
| 8. Oaxtepec | Morelos | 1534 | x | x | – | x | x | – | x | – | x | x | x | – | – | x | x |
| 9. Tepoztlán | Morelos | 1559 | x | x | – | x | x | – | – | – | – | x | x | – | – | x | x |
| 10. Ocuituco | Morelos | 1536 | x | – | x | x | – | – | x | x | – | x | – | x | x | – | – |
| 11. Zacualpan de Amilpas | Morelos | 1535 | x | – | x | x | – | – | – | – | – | x | – | x | – | – | x |
| 12. Tetela del volcán | Morelos | 1561 | x | x | x | – | x | – | – | – | – | x | – | x | – | x | x |
| 13. Tepeji del Río | Hidalgo | 1558 | x | – | – | x | x | – | – | – | – | x | x | – | x | x | – |
| 14. Alfajayucan | Hidalgo | 1559 | – | x | x | x | – | x | – | – | – | x | x | – | – | x | x |
| 15. Actopan open chapel | Hidalgo | 1550 | x | x | x | x | – | – | – | – | – | x | x | – | – | – | x |
| 16. Tezontepec | Hidalgo | 1554 | x | x | x | x | x | – | – | – | – | x | x | – | – | – | x |
| 17. Tepetitlán | Hidalgo | 1571 | – | x | – | x | x | – | x | – | x | – | – | – | – | x | – |
| 18. Actopan convent | Hidalgo | 1550 | x | x | x | x | – | – | – | – | – | x | – | – | – | – | – |
| 19. Atotonilco el grande | Hidalgo | 1536 | x | x | – | – | – | – | – | – | – | x | x | – | – | – | x |
| 20. Acatlán | Hidalgo | 1569 | x | x | – | x | – | – | – | – | – | x | x | x | – | – | – |
| 21. Yautepec | Morelos | 1550 | x | – | – | – | – | – | – | – | – | x | x | – | – | – | x |
| 22. Tlaltizapán | Morelos | 1576 | x | – | x | x– | x | – | x | – | – | x | x | x | – | – | x |
| 23. Tlaquiltenango | Morelos | 1590 | x | x | x | x | x | – | – | – | – | x | x | – | – | – | x |
| 24. Calpulalpan | Tlaxcala | 1576 | x | x | – | – | x | – | – | – | – | x | – | – | – | – | – |
| 25. Tepeapulco | Hidalgo | 1529 | x | x | x | x | – | – | – | – | – | x | – | x | – | x | x |
| 26. Zempoala | Hidalgo | 1569 | x | x | – | x | x | – | – | – | – | x | x | – | – | – | x |
| 27. Epazoyucan | Hidalgo | 1540 | x | – | x | x | x | – | – | – | – | x | x | – | – | – | x |
| 28. Oxoticpac | Estado de México | 1570 | x | – | x | x | x | – | – | – | – | x | x | – | – | – | x |
| 29. Cholula | Puebla | 1552 | x | x | x | x | x | – | x | – | – | – | x | x | – | x | – |
| 30. Huejotzingo | Puebla | 1530 | – | x | x | x | x | – | – | – | – | x | – | x | x | x | – |
| 31. Huatlatlauca | Puebla | 1566 | x | x | x | x | x | – | x | – | – | x | x | – | – | x | x |
| 32. Tecalli | Puebla | 1554 | x | x | – | x | x | – | x | – | – | x | – | x | – | x | |
| 33.Cuahutinchan | Puebla | 1554 | x | x | x | x * | x | – | – | – | x | x | x | – | – | x * | x |
| 34. Tula | Hidalgo | 1543 | x | x | – | x | x | – | – | – | – | x | x | – | – | – | – |
| 35. Meztitlán | Hidalgo | 1537 | x | x | – | x | – | – | – | – | – | x | x | – | – | x * | – |

st (stucco), ye (yellow), or(orange), bl (blue), bl * (greenish blue), br (brown), cr (cream), fl (flesh, gr (gray), pr (purple), bl (black), rd (red), bg (burgundy), pk (pink), gr (green), gr * (bluish green), pl (plaster). The colors that can be related to Maya blue are marked in bold and highlighted in gray.

X-ray fluorescence spectroscopy is a technique used to determine the elemental composition of materials. The technique is designed to detect the interactions of radiation with matter, i.e., the photoelectric effect of X-rays, which are characteristic of the elements that constitute a material [25]. The sample under analysis is first excited by a primary, high-energy X-ray; this causes the emission of characteristic X-rays of lower energy (fluorescence), which serve as a spectroscopic fingerprint for each element present in the sample.

By counting the number of photons within a certain emitted energy, the elements that compose the sample can be identified and quantified [26].

Portable XRF analysis is relevant to field research, as it allows for preliminary and non-invasive identification of materials, aiding in proper sample selection and reducing aggressive sampling. The portable XRF equipment has a compact system with a working voltage of 40 to 60 kV, with variable current in the micro amp range. It has been demonstrated that portable XRF analysis is an accurate and fast technique for the compositional characterization of a sample [27]. However, this type of equipment only allows for superficial examination of the sample, since the emitted energy is not high enough to penetrate subjacent layers. The scope of this type of equipment in the characterization of pictorial materials is limited, due to the fact that they often involve different layers and thicknesses. Therefore, the analysis of pictorial materials should be supplemented with other archaeometric techniques.

The XRF analysis was performed with an Innov-X Omega OSD2000 X-Ray handheld analyzer (Innov-X Systems, Inc., City, MA, USA), provided and available- by the Archaeology Laboratory of the Faculty of Social Sciences and Humanities at Universidad Autonoma de San Luis Potosí. The excitation source consists of an X-ray tube with large area Silicon Drift Detector (SDD). The SDD provides $10\times$ improvement in signal to background ratio, marked resolution improvement, and it operates at 195 to 165 eV (it has the capacity to handle $10\times$ more counts).

In this study, the analyzer window was placed directly in specific sections of the selected mural paintings in situ (according to the colors of interest) and each sample was irradiated for 2 min. The device was configured in MINING Application (PiN Detector: XPD6000, Innov-X Systems, Inc., City MA, USA) with a voltage of 40 kV and a current of 20 mA. This equipment detects up to 30 elements, with accuracy greater than 90% for calcium (Ca), iron (Fe), silicon (Si), magnesium (Mg) and aluminum (Al).

Before performing tests, it was necessary to standardize the instrument; thus, the analyzer hardware was initiated or restarted every time the instrument was operated for more than four hours. The procedure consists of placing the 316 stainless steel standardization clip on the analyzer nose and tapping the standardization (STD) function. It takes 20 s for the analyzer to calibrate to the parameters of the mode that has been selected for the tests. The automated standardization allows for the collection of a spectrum on a known standard (Alloy 316); it also involves the comparison of a variety of parameters to values stored when the instrument was calibrated at the factory.

The Innov-X Omega analyzer was set in MINING mode, in order to detect only metallic elements that might be present in the pigments. In addition, this analytical option helped to reduce the spectrum of the elements detected in the analysis and to characterize the typology of pigments, since the color hue of each pigment is given by a specific metal (iron, copper, manganese, etc.). Thus, this method allowed us to conduct an extensive preliminary in situ investigation of color and pigments, which then served to aid in our selection of locations and colors for physical sampling to further analyze them with SEM-EDS and XRD [28–30].

### 2.2. Scanning Electron Microscopy–Energy Dispersive X-ray Spectroscopy (SEM-EDS)

We selected 12 blue samples of 10 convents (Table 2) to be analyzed with a JEOL JSM6060LV (JEOL USA, Inc., Peabody, MA, USA) equipped with an INCA Energy 250 EDSLK-IE250 (Oxford Instruments, Oxfordshire, UK). The images were obtained on backscattering (BSE) mode in the surface and in the cross-sections of micro-fragments, under low vacuum, with 20 kV and diverse magnifications. Elemental composition was acquired using EDS in various modes: punctual, area and map distribution of chemical elements. The criteria described in Piovesan et al. [22] were followed to determine the painting technique.

**Table 2.** Number of blue samples that were taken from each convent, the location of the convent, their foundation year and their monastic affiliation.

| Convent | State | Year | Monastic Order | Number of Samples | Hue |
|---------|-------|------|----------------|-------------------|-----|
| Atlatlahucan | Morelos | 1569 | Augustinian | 1 | Greenish blue |
| Cuahutinchan | Puebla | 1554 | Franciscan | 2 | Light blue/dark blue |
| Cuernavaca | Morelos | 1552 | Franciscan | 1 | Blue |
| Ixmiquilpan | Hidalgo | 1550 | Augustinian | 1 | Blue |
| Meztitlán | Hidalgo | 1537 | Augustinian | 1 | Greenish blue |
| Oaxtepec | Morelos | 1534 | Dominican | 2 | Blue |
| Ocuituco | Morelos | 1536 | Augustinian | 1 | Blue |
| Tepeapulco | Hidalgo | 1529 | Franciscan | 1 | Blue |
| Tlaltizapán | Morelos | 1576 | Dominican | 1 | Blue |
| Tezontepec | Hidalgo | 1554 | Augustinian | 2 | Light blue/dark blue |

*2.3. X-ray Diffraction (XRD)*

Five samples previously analyzed with SEM-EDS were selected for X-ray diffraction (Table 3). These samples were chosen according to two different criteria: needle-shaped minerals were not clearly detected with SEM in the pigment layer; thus, it was important to ensure a clear identification of the mineral component of the blue pigment and/or the blue compositions showed relatively high quantities of Fe.

**Table 3.** Blue samples analyzed by XRD, showing a microscopic image and the location of the sample in each convent.

| Sample | Image with Optical Microscope | Photograph with the Blue Sample Location |
|--------|-------------------------------|------------------------------------------|
| Blue/Ocuituco (M1) |  |  |
| Light Blue/Tezontepec (M2) |  |  |
| Dark Blue/Tezontepec (M5) |  |  |
| Blur/Oaxtepec upper cloister (M3) |  |  |
| Blue/Oaxtepec lower cloister (M4) |  |  |

The samples sizes are very small and mainly consist of detached fragments due to conservation and displaying concerns. These wall paintings are taken from museums or convents still in use; thus, sampling should be restricted.

The samples were homogenized using an agate mortar and mounted on a zero-background sample holder due to their very small size.

X-ray diffraction spectra of non-oriented aliquots have been acquired using an Empyrean diffractometer (Malvern Panalytical, Malvern, UK), operating with an accelerating voltage of 45 kV and a filament current of 40 mA, and using CoKα radiation, nickel filter and a PIX-cel 3D detector (Malvern Panalytical, Malvern, UK). Samples were measured in the range of 4°–80° (2θ) with a step size of 0.002° (2θ) and 90 s of scan step time. Phase identification

and quantification by the Rietveld method [31] were completed using the Highscore v4.5 software (Malvern Panalytical, Malvern, UK), as well as ICCD (International Center for Diffraction Data) and ICSD (Inorganic Crystal structure Database) databases.

In a second stage, the samples were heated to 450 °C and measured by X-ray diffraction under the same conditions as the original samples, for the purpose of verifying the destruction of the basal peaks of palygorskite (~10.3 Å) and sepiolite (~12 Å) and, in turn, confirming the presence of these mineral phases [32].

## 3. Results

The next paragraphs show the results obtained with each technique. Overall, results showed that Maya blue could have been widespread in XVI century wall painting in convents.

### 3.1. XRF. Concentration of Selected Elements in Blue Pigments

Metal elemental compositions of blues were compared with the metal elemental composition of stucco where possible (see Appendix B, where all the samples and analyzed elements are shown); Huejotzingo, Tepetitlán and Alfajayucan had a white painting coating over almost every space of stucco, preventing the acquisition of high-quality information.

Some blues used in colonial times (see Introduction) and other pigments added in the XIX and XX centuries could be found in the convents' wall paintings, all of them having a distinct composition. Thus, XRF analysis allowed us to perform a quick study of the composition of blues and, consequently, to devise a hypothesis of the most likely used pigment.

Regarding the amount of iron (Fe) and the possibility of Prussian Blue repaints, results show that 12 hues of blues have a greater increase in Fe compared with the quantity in the stuccos, 14 hues of blues have a slight increase in Fe, while 5 have almost the same amount and 3 have less Fe in comparison to the corresponding stucco.

Copper (Cu) was found only in three samples (the samples from Tepeji del Río and one sample from Cholula) in ranges between 0.01% and 0.034% while the corresponding stucco had no Cu (Figure 2). It is possible that the blues found in Tepeji are made of a copper-based pigment while the sample from Cholula (0.01% of Cu) could be a mixture of blue pigments including a copper-based one.

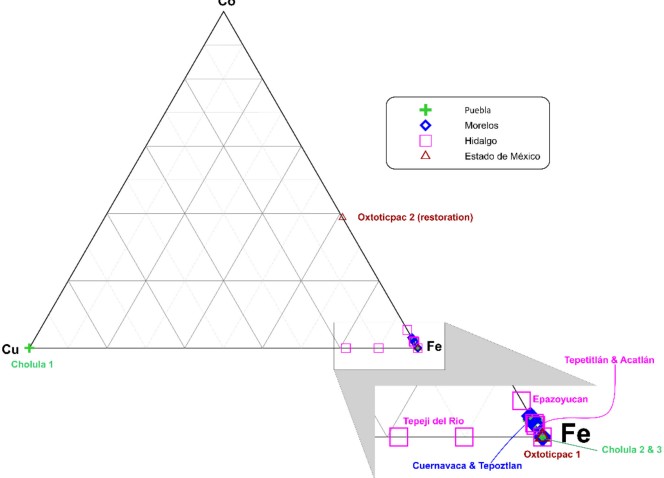

**Figure 2.** Fe–Cu–Co triangular diagram of the studied blue pigments from convents of different states in Mexico. Data acquired with EDS.

Regarding cobalt (Co), two blue samples (Atlixco and Zacualpan de Amilpas) have Co in their compositions. However, the stucco had similar quantities of this element; hence, it is not probable that the blue sample is a cobalt-based blue. There were, nonetheless, other

samples that exhibited Co while their corresponding stuccos did not. These samples were one sample from Cuernavaca, the sample from Tepoztlan (which also exhibited a high quantity of Fe), one sample from Tepetitlán, the sample from Acatlan, one sample from Epazoyucan and the sample from Oxtoticpac, clearly recognized as a restoration. These pigments could be cobalt-based pigments, and as such, they could correspond to later (XVIII or XIX century) repainting or to restoration interventions. The sample of Tepoztlán could be a mixture of Prussian blue and a cobalt-based pigment.

The sample from Oxtoticpac mentioned above (a clear restoration intervention), as well as the analyzed areas of Tepeji del Río, showed clear differences between the global composition when compared with the other blues.

### 3.2. SEM-EDS. Painting Layer Micromorphology

It can be noticed in all the scanning electron microscopy images (Figures 3 and 4) that the color is clearly in a different layer than the stucco and plaster; therefore, none of these blues were applied using a fresco technique. All were painted with a dry technique using an organic binder (see Table 4), except for the Tezontepec and Cuernavaca samples, where the pigment appears to be mixed with gypsum and lime in a technique similar to lime painting but also incorporating calcium sulfate.

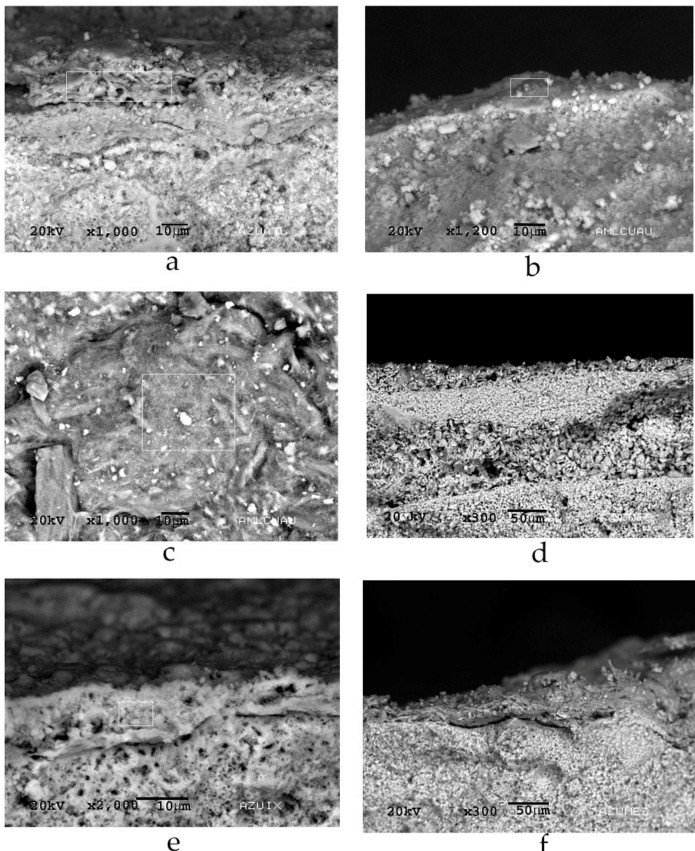

**Figure 3.** SEM images of different blue samples. (**a**) Atlatlaucan cross-section with three main layers. The one at the top is the paint layer that is placed over a stucco and, on the bottom, the plaster can be seen. (**b**) Cuauhtinchan, cross-section with three main layers. The one at the top is the paint layer that is placed over a thin white layer of stucco placed over a plaster. (**c**) Cuauhtinchan, surface of the pain layer, a mesh of thin needle-shaped features can be seen. (**d**) Cuernavaca cross-section. The pigment is dispersed within the surface layer, where dark areas can be observed. The pictorial layer is placed over several layers of gypsum. (**e**) Ixmiquilpan, cross-section. The top layer is the blue paint placed over a thin white layer made of gypsum, and on the bottom, the plaster is observed. (**f**) Meztitlán, cross-section. The paint layer is placed directly over a plaster.

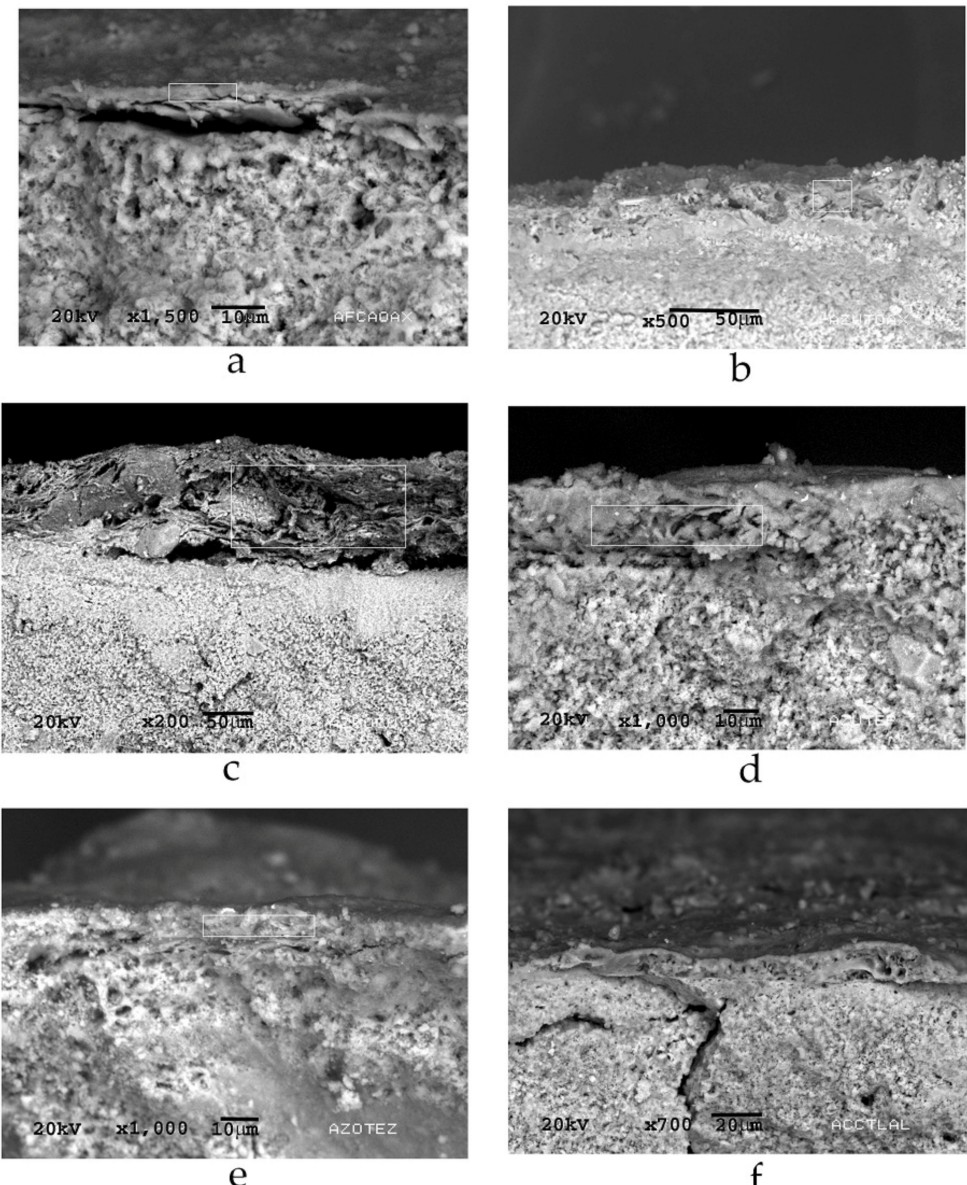

**Figure 4.** SEM images of different blue samples. (**a**) Oaxtepec upper cloister, cross-section with three main layers. The one at the top is the paint layer that is placed over a stucco barely distinguishable from the plaster by its coarser and more porous texture. (**b**) Oaxtepec lower cloister, cross-section. The dark layer on top is the blue paint placed over a stucco, and on the bottom, a less porous plaster is observed. (**c**) Ocuituco, cross-section. The blue paint layer is thick and placed over a compact stucco layer; at the bottom, a more porous plaster is observed. (**d**) Tepeapulco, cross-section. A thin dark layer of blue paint is placed over a stucco; then, the more compact plaster is seen. (**e**) Tezontepec, cross-section. A thin dark layer of blue paint is placed over a thin stucco that is clearly separated from the plaster on the bottom. (**f**) Tlaltizapan, cross-section. The paint layer is directly placed over the plaster.

**Table 4.** Elemental analysis of blue samples with EDS. The elements and percentage are shown in the columns. Ca and S concentrations are related to the presence of gypsum that was identified with XRD. The high concentration of C is related to the use of an organic binder in the paint.

| Convent | Sample | C | O | Na | Mg | Al | Si | P | S | Cl | K | Ca | Fe | Cu | Pb | Total |
|---|---|---|---|---|---|---|---|---|---|---|---|---|---|---|---|---|
| Atlatlauhcan | Greenish blue | **22.24** | 48.24 | – | 0.62 | 1.35 | 5.03 | – | **5** | 0.32 | 0.32 | 16.14 | 0.74 | – | – | 100 |
| Cuauhtinchan | Blue | **19.59** | 49.54 | – | 1.55 | 2.24 | 6.15 | – | **6.13** | – | – | 14.79 | – | – | – | 100 |
| Cuauhtinchan | Blue lateral wall | **12.61** | 51.62 | – | 8.88 | 0.42 | 19.43 | – | **0.77** | 0.15 | – | 4.71 | 1.09 | 0.31 | – | 100 |
| Cuernavaca | Blue | **42.61** | 41.98 | – | - | 0.23 | 0.74 | – | **2.71** | 11.73 | – | – | – | – | – | 100 |
| Ixmiquilpan | Blue | **16.04** | 45.17 | – | 0.25 | 0.16 | 1.39 | – | **4.2** | 0.25 | – | 31.68 | – | – | 0.85 | 100 |
| Meztitlán | Blue | **24.07** | 46.78 | – | 1.34 | 1.34 | 9.37 | **2.15** | **1.65** | 0.13 | **0.29** | 12.57 | 0.31 | – | – | 100 |
| Oaxtepec | Blue upper cloister | **29.5** | 50.53 | **0.14** | 0.41 | 0.63 | 3.31 | **0.49** | **3.31** | 0.36 | **0.31** | 10.87 | 0.14 | – | – | 100 |
| Oaxtepec | Blue lower cloister | **27.12** | 46.42 | – | 1.11 | 0.89 | 5.567 | **0.42** | **1.15** | – | **0.59** | 16.09 | 0.54 | – | – | 100 |
| Ocuituco | Blue | **18.5** | 55.89 | – | 1.96 | 0.35 | 4.8 | – | **5.3** | – | **0.3** | 12.9 | – | – | – | 100 |
| Tepeapulco | Blue | **22.51** | 50.96 | – | 2.79 | 1.19 | 8.24 | – | **1.57** | 0.34 | **0.24** | 11.73 | 0.42 | – | – | 100 |
| Tezontepec | Dark blue | **45.54** | 42.53 | **0.34** | 0.19 | 0.51 | 1.17 | – | **3.4** | 0.19 | **0.06** | 5.89 | 0.18 | – | – | 100 |
| Tezontepec | Light blue | **52.27** | 35.93 | **0.21** | 0.2 | 0.15 | 0.63 | – | **3.77** | – | – | 6.64 | 0.2 | – | – | 100 |
| Tlaltizapan | Blue | **25.47** | 49.62 | **0.51** | 0.96 | 0.8 | 9.93 | **0.36** | **0.89** | 0.26 | **0.51** | 10.44 | 0.25 | – | – | 100 |
| – | Egg (mixed white and yolk) | **85.67** | 12.49 | **0.16** | – | – | – | **0.54** | **0.57** | 0.36 | **0.21** | – | – | – | – | 100 |

Elements related to palygorskite and sepiolite are highlighted with grey shading, while elements possibly related to an organic binder are marked in bold.

All samples have a clear stucco or preparation base (a thin layer between the plaster and the colour layer) below the painting layer, with the exception of the samples from Meztitlán and Tlaltizapan. In the first one, the blue color is placed directly on a lime plaster, while in the latter, a gypsum–lime plaster is the base. In the other samples, the blue colors are placed on a gypsum–lime stucco or preparation base or on stucco made only from gypsum; this is clearly a technique introduced by Spaniards considering that, to date, there is almost no scientific information that indicates the use of gypsum in wall painting in pre-Hispanic times. In Figure 3c, this corresponds to the blue lateral wall sample from Cuauhtinchan, in which a net of a mineral with a needle-shaped morphology corresponding to the palygorskite can clearly be seen.

On the other hand, to estimate the elemental composition of the pictorial layer, the samples were analyzed in cross-sections with EDS and focused area analyses were carried out only at this layer; the complete results are shown in Table 4. In all studied areas of all the samples, silica (Si), aluminum (Al) and magnesium (Mg) elements were found, and these elements correspond to the composition of palygorskite, except for the sample from Cuernavaca, in which no traces of Mg or Fe were found. Several of them also present iron in their composition.

On the SEM images of the samples, some physical deterioration can be observed, including fissures, fractures and detachments of the paintings. The sampling method, in which some pressure had to be applied to separate the minimal samples, could have caused some of these features. Nonetheless, major chemical changes are not expected in the mineral fractions of the paint layer, the stucco or plasters of the studied samples because no morphological evidence of dissolution or alteration of original substances was found, such as etch pits and synthesis of neoformed components as secondary carbonates. Usually, such components have quite specific shapes, showing a sharp difference from the original constituents of stucco and pigment layers. We also did not find any biological agents of deterioration—cells of cianobacteria, fungi, etc. This is partly due to the fact that for the SEM/EDX analysis, we selected fragments with minimal macroscopic deterioration features. Thus, supposedly, the chemical signal from mineral components from EDS microanalysis originates predominantly from the primary components and is not influenced significantly by posterior alteration.

To identify unequivocally these minerals, it would be ideal to calculate their structural formula from chemical point analysis obtained with a scanning electron microprobe. However, given the porous nature of the material and the small amount of pigment available, this type of analysis was not possible, and we had to use SEM-EDS data to be able to chemically classify them without calculating their structural formula.

It is important to mention that published results of microanalyses of individual particles of both sepiolite and palygorskite are quite rare; this is because these results can be affected by other clay minerals and other associated minerals, such as impurities [33]. Many of the palygorskite samples contain impurities and, thus, obtaining an accurate analysis is always difficult. This is why it was decided to use the triangular diagram (Figure 5) of the AFM type ($Al_2O_3$–$Fe_2O_3$–MgO) to compare the analyzed samples (SEM-EDS) of blue pigments from the different convents with each other—and mainly with the palygorskite and sepiolite reported in the mineralogical literature [33]. Some palygorskite samples reported in Maya blue-type pigments from the Mayan zone were also included [34].

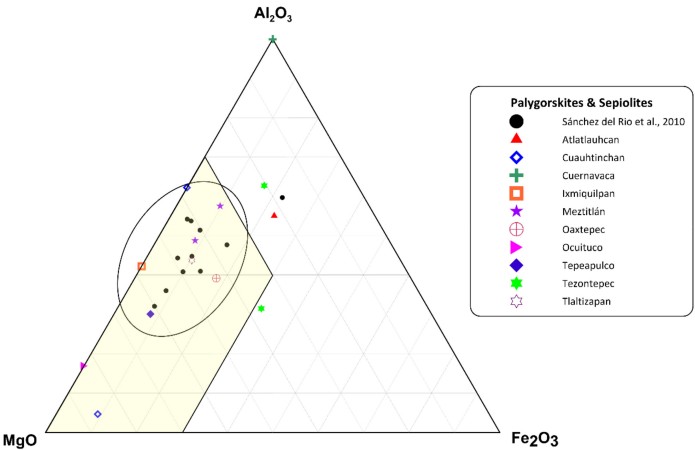

**Figure 5.** Triangular AFM diagram. In yellow, area determined for palygorskite and sepiolite samples studied by García-Romero & Suárez [33].

With the exception of the samples from Atlatlahucan (slightly enriched in aluminum), Tezontepec (slightly enriched in aluminum and iron) and Cuernavaca (without iron and magnesium), all other samples of blue pigment are projected within the field established for the sepiolite–palygorskite group [33]. Additionally, most of them—with the exception of a sample from Cuauhtinchan and Ocuituco—are very close to the samples reported by Sánchez del Río et al. [34], for Maya blue-type pigments collected in the area of the Yucatán Peninsula. These last two pigment samples could correspond to sepiolite, since this mineral occupies the most magnesic and trioctahedral extreme, while palygorskite occupies the most aluminic–magnesic and dioctahedral extreme and has a lot of vacancies. It is important to mention that, unlike what many authors have indicated in the past, these authors [33] show that there is no compositional gap between both mineralogical species.

Moreover, the phosphorus, potassium, chlorine and sodium found with EDS can correspond to some elements present in the binders mixed with the pigments. Collagen and egg, which were commonly used to paint in secco techniques, usually present these elements. In Figure 6, we compare the elemental composition of egg (mixture of whites and yolk) and some blue samples.

As it can be observed in the graph, the elemental composition matches well with a proteic binder, such as egg; nevertheless, further analyses should be carried out to determine the exact nature of the binder.

Sulfur is high in the blue samples and this aspect can be related to the preparation of stuccos or preparation bases, which contain or are completely made of gypsum. In addition, gypsum was added as a binder in the painting layer in some cases. Phosphorus is also high if compared with the quantity present in the egg in the Meztitlán sample. This sample has a thin, black layer in some areas over the blue. Thus, this black could be made from ash or carbonized bone.

The samples from Cuernavaca also exhibit a high quantity of chlorine, which can possibly be correlated to the presence of salts within the wall paintings. In one of these samples

from Cuernavaca, a sodium nitrate salt was found (Figure 7), indicating a conservation problem caused by saline efflorescence.

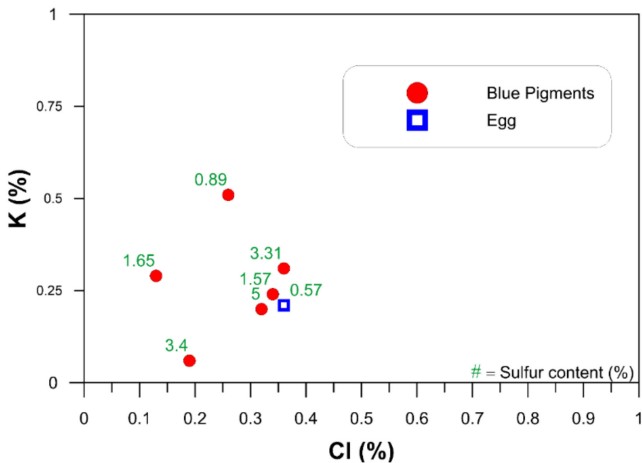

**Figure 6.** Comparison between elemental composition (K, Cl, S) of blue samples and hen egg (yolk and white, mixed).

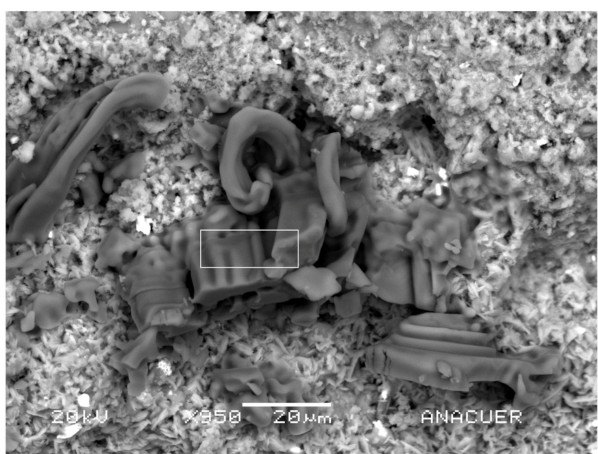

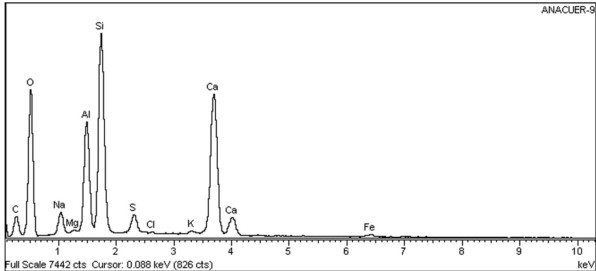

**Figure 7.** Efflorescence and their elemental composition found in a sample of Cuernavaca.

### 3.3. XRD. Mineralogical Composition of the Pigments

Five samples (M1 to M5) were characterized. By means of X-ray diffraction analysis, the presence of sepiolite in one sample (M1), and of palygorskite in the other four samples (M2, M3, M4 and M5), could be confirmed. The mineralogical results are shown in Table 5. The identification of the sepiolite was based on the presence of peaks with interplanar distance of 11.9 and 4.45 Å. The identification of palygorskite was carried out based on the presence of peaks with interplanar distance of 10.36 and 4.46 Å (Figure 8).

**Table 5.** XRD results. Quantitative results by Rietveld method could only be obtained for samples M1 and M5. GOF = goodness of fit. IMA (International Mineralogical Association) approved formulas of minerals (http://cnmnc.main.jp/) are indicated.

| Sample | Identified Phases | XRD Patterns | Quantitative Rietveld (%) | GOF |
|---|---|---|---|---|
| M1 Blue/Ocuituco | Calcite: $CaCO_3$ <br> Gypsum: $CaSO_4 \cdot 2H_2O$ <br> Bassanite: $CaSO_4 \cdot 0.5H_2O$ <br> Sepiolite: $Mg_4Si_6O_{15}(OH)_2 \cdot 6H_2O$ | ICDD 01 072 4582 <br> ICSD 98 016 1623 <br> ICDD 01 074 2787 <br> ICDD 98 003 1142 | 32.6 (7) <br> 36.9 (5) <br> 8.6 (9) <br> 21.9 (9) | 0.967 |
| M2 Light Blue/ Tezontepec | Calcite: $CaCO_3$ <br> Gypsum: $CaSO_4 \cdot 2H_2O$ <br> Palygorskite: $(Mg,Al)_2Si_4O_{10}(OH) \cdot 4H_2O$ | ICDD 01 072 4582 <br> ICSD 98 016 1623 <br> ICDD 98 004 0688 | – | – |
| M3 Blue/Oaxtepec upper cloister | Calcite: $CaCO_3$ <br> Gypsum: $CaSO_4 \cdot 2H_2O$ <br> Palygorskite: $(Mg,Al)_2Si_4O_{10}(OH) \cdot 4H_2O$ | ICDD 01 072 4582 <br> ICSD 98 016 1623 <br> ICDD 98 004 0688 | – | – |
| M4 Blue/Oaxtepec lower cloister | Calcite: $CaCO_3$ <br> Palygorskite: $(Mg,Al)_2Si_4O_{10}(OH) \cdot 4H_2O$ | ICDD 01 072 4582 <br> ICDD 98 004 0688 | – | – |
| M5 Dark Blue/ Tezontepec | Calcite: $CaCO_3$ <br> Gypsum: $CaSO_4 \cdot 2H_2O$ <br> Bassanite: $CaSO_4 \cdot 0.5H_2O$ <br> Palygorskite: $(Mg,Al)_2Si_4O_{10}(OH) \cdot 4H_2O$ | ICDD 01 072 4582 <br> ICSD 98 016 1623 <br> ICDD 01 074 2787 <br> ICDD 98 004 0688 | 43.2 (8) <br> 33.1 (9) <br> 10.4 (7) <br> 13.2 (8) | 0.953 |

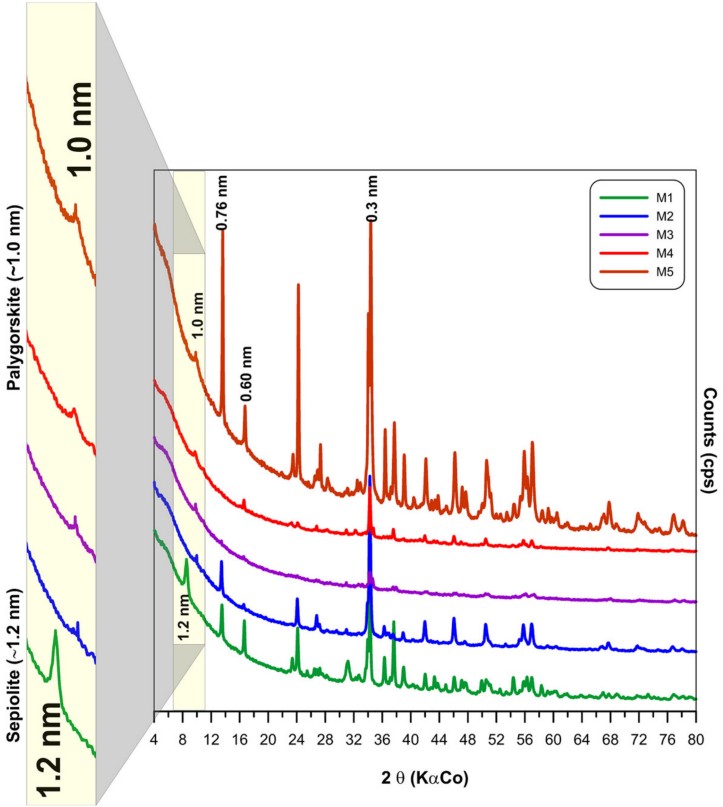

**Figure 8.** XRD patterns of the five measured samples (M1, M2, M3, M4 and M5). The samples were measured using a fine focus cobalt tube (KαCo radiation). The basal peaks with interplanar distance of 1.0 and 1.2 nm enlarged at the left of the figure are characteristic of palygorskite and sepiolite, respectively.

Paligorskite and sepiolite are minority phases in all samples due to the presence of abundant gypsum, bassanite and non-magnesian calcite in the paste, showing that the blue pigmented layer is very thin (Table 5). Quantification by the Rietveld method (Table 5) could only be applied to samples M1 and M5 because the other samples were very small, and the signal-to-noise ratio does not allow us to obtain good fit indices. Figure 9 shows the result of the Rietveld refinement for samples 1 and 5.

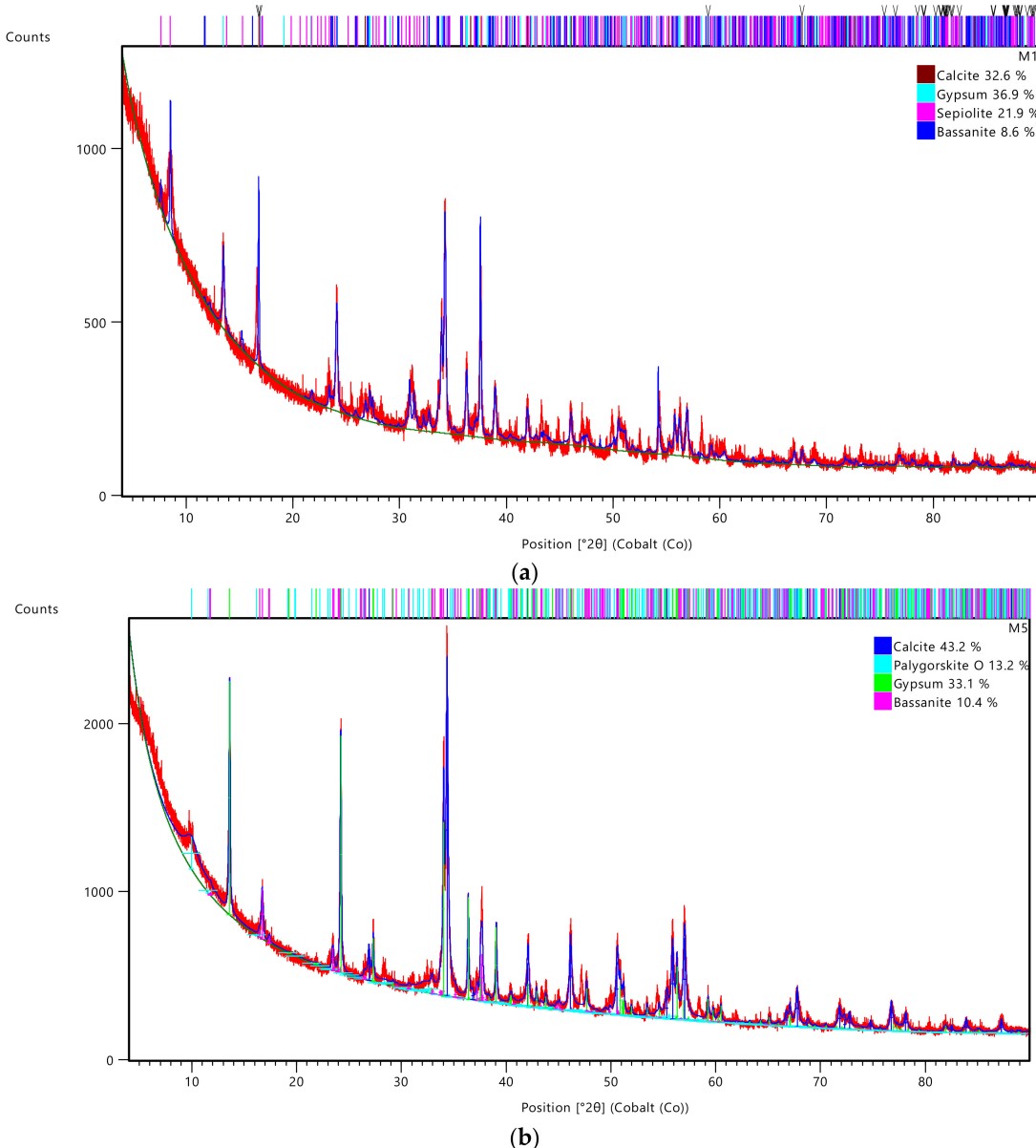

**Figure 9.** Quantitative results obtained by Rietveld refinement method; (**a**) sample M1 (GOF = 0.967); (**b**) sample M5 (GOF = 0.953); GOF = Goodness of fit.

The cell parameters obtained from the Rietveld refinement of M5 palygorskite are: **a** = 17.86, **b** = 5.21 Å, **c** = 12.77 Å, **α** = **β** = **γ** = 90°. These data match very well with those of the diffraction pattern (ICSD 98-004-0688) used to identify this mineral (Table 6).

**Table 6.** Cell parameters for sepiolite (sample M1) and palygorskite (sample M5) calculated by Rietveld refinement method.

| Sample | Mineral | a | b | c | α | β | γ | V(Å³) |
|--------|---------|-----|-----|-----|-----|-----|-----|-------|
| Blue/Ocuituco (M1) | Sepiolite | 5.28(5) | 13.40(6) | 26.80 (1) | 90° | 90° | 90° | 1898 |
| Dark blue/Oaxtepec (M5) | Palygorskite | 17.86 (9) | 5.21(6) | 12.77(4) | 90° | 90° | 90° | 1189 |

The cell parameters obtained from the Rietveld refinement of M1 sepiolite are: **a** = 5.28 Å, **b** = 13.40 Å, **c** = 26.80 Å, **α** = **β** = **γ** = 90°. These data match very well with those of the diffraction pattern (ICSD 98-003 1142) used to identify this mineral (Table 6).

## 4. Discussion

### 4.1. Chemical and Mineralogical Identification of Maya Blue

The majority of the samples analyzed with XRF fulfill the elemental criteria to be composed of Maya blue. Al of them have Si, Al and Mg.

In addition, when the elemental composition of XRF and the mineral composition of the blue samples are put together and compared (see Appendix A, where a table with direct comparison can be found), it can be observed that, even when the samples contained high values of Fe compared with those of the stucco, palygorskite was found. This is, as mentioned above, an almost certain indication that the blues were made with Maya blue, as shown in samples from Oaxtepec and Tezontepec, for example. Thus, high values of Fe acquired with non-destructive techniques should not be immediately attributed to the presence of Prussian blue. Furthermore, on the ubiquitous presence of Fe in these blue samples, several analyses of Maya blue suggest that Fe in the pigment replaces Mg and/or Al ions in the octahedral layer of the palygorskite, which occurs in all natural palygorskites [11]. However, as the pictorial layers and the stucco layers are so thin, XRF could be acquiring data from the plaster layer, and some quantity of the Fe could be due to the aggregates in the mixture with binders.

Nevertheless, other analyses had to be completed to confirm the presence of Maya blue conclusively. Using SEM-EDS, it was observed that all samples had the correct elemental composition and micromorphology to have a palygorskite-based pigment and thus the probability of these blue samples containing Maya blue.

Furthermore, samples studied by XRD exhibited palygorskite or sepiolite, which is a solid and unmistakable confirmation that Maya blue was used in the elaboration of the blue pigments employed in the wall paintings of these convents. Of the five pigment samples studied, four are characterized by the presence of palygorskite and one (M1) by the presence of sepiolite. In no sample have both minerals been found mixed (Figure 10).

Several palygorskites of great purity have been collected in several regions of the Yucatán Peninsula, all located in a radius of 40 km around the archaeological site of Uxmal [34]. The genesis of these deposits has been studied by several authors [35,36], and the most accepted interpretation in relation to their origin is that they were formed by direct crystallization in saline lagoons and on the shallow sea floor of the Yucatán Platform. The palygorskite–sepiolite clays of the Yucatan Peninsula are interbedded with limestone and dolomites.

The most reported mineral in the Yucatán Peninsula is palygorskite [36,37]; however, the presence of sepiolite has also been mentioned by multiple authors [35,38], but it has never been described in the deposits that emerge on the surface—those that were used by the Mayans in the manufacture of the Maya blue pigment. This is why sepiolite has never been identified in archaeological Maya blue to date and why the manufacture of this pigment has been exclusively associated with the mineral palygorskite [34,36,37,39–42]. However, it is very interesting to mention that sepiolite was found in archaeological samples from The Great Temple in Tenochtitlan, corresponding to the Aztec Empire [43,44].

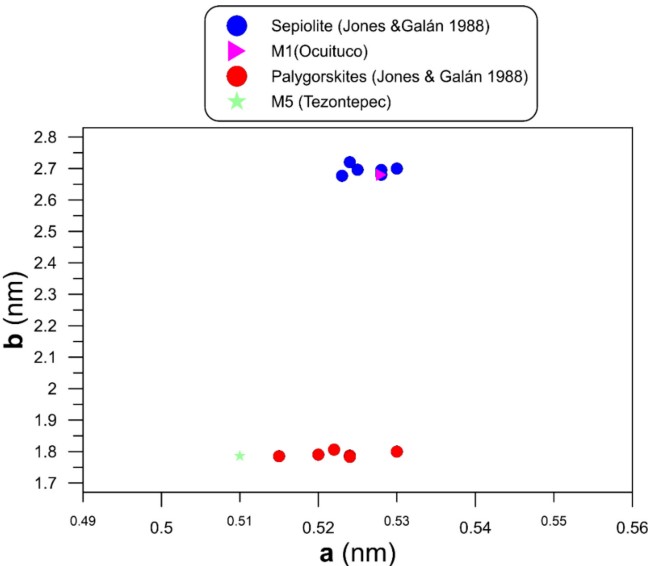

**Figure 10.** Cell parameters (**a,b**) for sepiolite of sample M1 (Ocuituco) and palygorskite of sample M5 (Tezontepec). Comparison with the cell parameters of the same minerals reported by Jones and Galán [45].

The mineralogical and elemental composition of the samples suggests that the palygorskite found in the blue samples of Oaxtepec and Tezontepec came from the Yucatan Peninsula and thus from the Mayan area; this evidence poses an interesting question about trading with a region that was not completely conquered in the XVI century. The sample from Ocuituco in which sepiolite was found has, with no doubt, another source that was not identified in this paper.

Finally, we want to emphasize the absence of pigments such as azurite or smalt. We know that their exclusion was not due to the lack of materials, since there was an active market for pigments from the European continent to New Spain and vice versa.

In the case of azurite from 1550 onwards, this blue pigment was a constant merchandise in the commercial trade between the Indies and Seville, and it was developed during the entire XVI century and a great part of the XVII century; it was extracted from mines in Santo Domingo and shipped in large quantities to Spain [20]. Moreover, azurite has been found widely in New Spain's oil canvases and wooden paintings [19]; its price varied between 34 and 54 maravedíes per pound [46].

On the other hand, in regard to blue smalt, Nicolás de Lambartengo, a neighbor of Seville, sent "*5 libras de esmalte de vidrio a 6 reales la libra, 1020 maravedíes*" in the ship Nuestra Señora de la Victoria in 1586 [46]. This report reveals small but constant quantities that were shipped, as well as their high cost, since at the time, a turkey cost approximately one real [47]; that is, to buy one pound, six turkeys would have been necessary. After azurite, blue smalt was the pigment most used by artists from New Spain, sometimes even mixed with azurite to achieve grayish shades [19].

In the case of mural painting, the quantities of pigments that were needed kept the Maya blue trade active. By maintaining the use of this pigment, not only were they using a specific material and tonality, but they kept the notions and meaning of this particular blue alive, becoming a suitable element for the integration of the indigenous culture to the western tradition.

*4.2. Possible Interventions in the Wall Paintings*

Samples from Cuernavaca, Tepoztlan, Tepetitlán, Acatlan, Epazoyucan, Oxtoticpac and Tepeji del Río probably had other pigments with/or instead of Maya blue, as shown by XRF analyses. The elemental composition showed elements such as cobalt or copper, which correspond to other blue pigments. These could be found for many reasons: (1) wall

paintings were refreshed and re-painted later in the XVIII/XIX century; (2) wall paintings were possibly restored and have been subject to non-detectable (by eye) interventions; or (3) they have been restored in a way that can be clearly seen, as in Oxtoticpac.

In addition, when the structural formula of the clay minerals (palygorskite and sepiolite) was being calculated, an unusually high quantity of $SiO_2$ was noticed when compared with the quantity reported in the literature. For example, in the case of the blue sample of Tlaltizapan (Figure 11), it can be seen (both in spectra and in the image where a dark coating is present) that the surface is highly enriched with Si. The dark color points towards a lighter elemental composition than the rest of the material, reflecting the Si composition against the calcium carbonate and gypsum composition of the other layers.

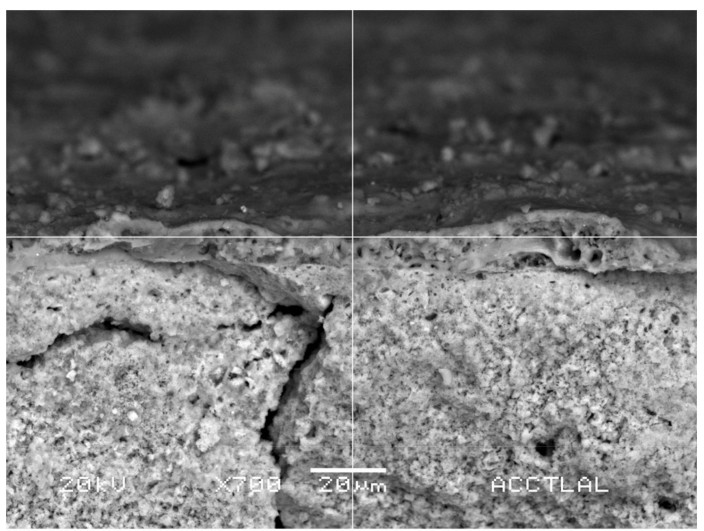

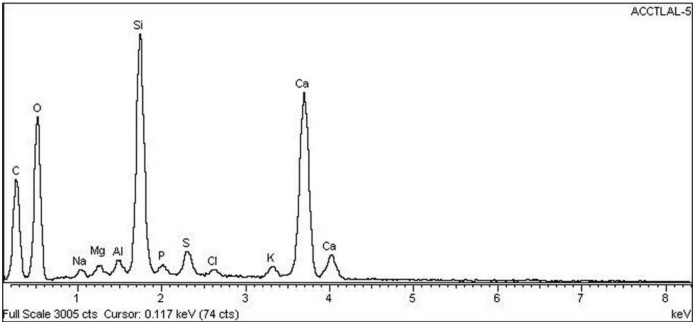

**Figure 11.** SEM image of Tlaltizapan blue. A dark coating enriched in Si can be observed at the surface.

We can explain this in two ways: firstly, that the wall paintings were submitted to a silicate-based consolidation process; and second, that in a given time, the wall paintings were subjected to a waterproofing treatment.

## 5. Conclusions

To conclude, Maya blue was apparently the preferred blue color for wall painting during the early and late XVI century, extending the use and life of a pre-Hispanic knowledge to the western Catholic tradition. It is noteworthy that none of the samples were painted al fresco, but with a variety of secco techniques. Moreover, almost every sample had a thin stucco or preparation base made of gypsum, a technique that was probably imported from Spain because the pre-Hispanic wall painting technique was almost always carried out with lime plasters and stuccos.

It is remarkable that the composition of palygorskite found in this research matches the compositions of those studied in the Yucatán Peninsula; this information indicates an

active trade between different regions. In addition, it is important to highlight the presence of sepiolite in the blue sample from Ocuituco, as it corresponds to an entirely different tradition of pigments and perhaps even different trade routes, since there has not been any sepiolite found in Maya blues from the Mayan area.

**Author Contributions:** Conceptualization L.S.-M. and T.P.-P.; methodology, L.S.-M. and T.P.-P., A.F.-M., B.L.-T., K.C.-C., and M.F.U.-L.; investigation L.S.-M., T.P.-P., A.F.-M., B.L.-T., K.C.-C., and M.F.U.-L., resources, P.E.-G., A.F.-M., L.S.-M., and S.S.; data curation, L.S.-M. and T.P.-P.; writing—original draft preparation, L.S.-M., T.P.-P., A.F.-M., K.C.-C., and B.L.-T.; writing—review and editing, L.S.-M. and T.P.-P.; funding acquisition P.E.-G., A.F.-M., L.S.-M., and S.S. All authors have read and agreed to the published version of the manuscript.

**Funding:** This research was funded by PROGRAMA DE APOYO A PROYECTOS DE INVESTIGACIÓN E INNOVACIÓN TECNOLÓGICA (PAPIIT-UNAM) No. IN401918 and CONACYT: Laboratorios Nacionales, No. 299087.

**Data Availability Statement:** Data sharing not applicable.

**Acknowledgments:** The authors thank Mireia Lilit Solé Pi by the final English revision of the manuscript.

**Conflicts of Interest:** The authors declare no conflict of interest.

## Appendix A

Convents with wall paintings are: Acatlán (Hidalgo), Acolman (Edo. Méx.), Actopan (Hidalgo), Alfajayucan (Hidalgo), Altihuetzia (Tlaxcala), Amecameca (Edo. Méx.), Apan (Hidalgo), Atlatlahucan (Morelos), Atlixco (Puebla), Atotonilco el Grande (Hidalgo), Ayotzingo (Edo. Méx.), Azcapotzalco (Cd. Mx.), Calpulalpan (Tlaxcala), Chimalhuacán-Chalco (Edo. Méx.), Cholula (San Gabriel) (Puebla), Churubusco (Cd. Mx.), Coyoacán (Cd. Mx.), Cuautinchán (Puebla), Cuernavaca (Morelos), Culhuacán (Cd. Mx.), Epazoyucan (Hidalgo), Huatlatlauca (Puebla), Huejotzingo (Puebla), Huexotla (Edo. Méx.), Ixmiquilpan (Hidalgo), Ixtapaluca (Edo. Méx.), Iztacalco (Cd. Mx.), Izúcar (Puebla), Juchitepec (Edo. Méx.), Malinalco(Edo. Méx.), Metepec (Edo. Méx.) Metztitlán (Hidalgo), Milpa Alta (Cd. Mx.), Molango (Hidalgo), Oaxtepec (Morelos), Ocuituco (Morelos), Oxtoticpac (Estado de México), San Jacinto Tenanitla (San Ángel) (Cd. Mx.), Singuilucan (Hidalgo), Tecali (Puebla), Tehuacán (Puebla), Tepeapulco (Hidalgo), Tepeji del Río (Hidalgo), Tepetitlán (Hidalgo), Tepetlaoxtoc (Edo. Méx.), Tepeyanco (Tlaxcala), Tepoztlán (Morelos), Tequixquiac (Hidalgo), Tetela del Volcán (Morelos), Tezontepec (Hidalgo), Tláhuac (Cd. Mx.), Tlahuelilpa (Hidalgo), Tlalmanalco (Edo. Méx.), Tlalnepantla (Edo. Méx.), Tlaltizapán (Morelos), Tlaquiltenango (Morelos), Tlaxcala (Tlaxcala), Tlayacapan (Mor.), Tochimilco (Puebla), Tula (Hidalgo), Xochimilco (Cd. Mx.), Yautepec (Morelos), Yecapixtla (Morelos), Zacatlán de las manzanas (Puebla), Zacualpan Amilpas (Morelos), Zempoala (Hidalgo) y Zinacantepec (Edo. Méx.) [48].

## Appendix B

Elemental analysis of blue samples with XRF. The elements and percentage are shown in the columns and LE refers to light elements. Only the quantity of metals was intentionally tracked with XRF. The blue samples marked with * have lower quantities of Fe than stucco, samples marked with ~ had almost the same amount of Fe than stucco, the ones that are marked with ** have slightly more amount of Fe, and the blue samples marked with *** have higher amounts of Fe than the corresponding stucco.

**Table A1.** Elemental analysis obtained with XRF.

| Convent | State | Hue | Ti | Mn | Fe | Co | Zr | Zn | Pb | V | Sn | Sb | Bi | Cr | Cu | Ni | As | LE |
|---|---|---|---|---|---|---|---|---|---|---|---|---|---|---|---|---|---|---|
| Yecapixtla | Morelos | Blue *** | 0.019 | 0.018 | 0.404 | – | – | – | – | – | – | – | – | – | – | – | – | 99.547 |
| Yecapixtla | Morelos | Stucco | 0.029 | – | 0.189 | – | – | – | – | – | – | – | – | – | – | – | – | 99.761 |
| Atlixco | Puebla | Blue *** | 0.044 | 0.01 | 0.999 | 0.023 | 0.012 | – | – | – | – | – | – | – | – | – | – | 98.927 |
| Atlixco | Puebla | Stucco | 0.022 | – | 0.553 | 0.015 | 0.01 | – | – | – | – | – | – | – | – | – | – | 99.394 |
| Ixmiquilpan | Hidalgo | Blue ** | 0.034 | – | 0.316 | – | – | – | 0.017 | – | – | – | – | – | – | – | – | 99.615 |
| Ixmiquilpan | Hidalgo | Stucco | 0.037 | – | 0.246 | – | – | – | – | – | – | – | – | – | – | – | – | 99.696 |
| Cuernavaca | Morelos | light blue *** | 0.026 | – | 0.515 | – | 0.015 | 0.102 | 0.018 | – | – | – | – | – | – | – | – | 99.306 |
| Cuernavaca | Morelos | dark blue *** | 0.079 | – | 0.341 | 0.011 | 0.014 | – | 0.012 | 0.028 | – | – | – | – | – | – | – | 99.498 |
| Cuernavaca | Morelos | light blue *** | 0.026 | – | 0.515 | – | 0.015 | 0.102 | 0.018 | – | – | – | – | – | – | – | – | 99.306 |
| Cuernavaca | Morelos | Stucco | 0.056 | – | – | – | – | – | – | – | – | – | – | – | – | – | – | 99.927 |
| Oaxtepec | Morelos | blue/uper cloister~ | 0.024 | – | 0.077 | – | – | – | – | – | – | – | – | – | – | – | – | 99.881 |
| Oaxtepec | Morelos | Stucco | 0.012 | – | 0.079 | – | – | – | – | – | – | – | – | – | – | – | – | 99.893 |
| Oaxtepec | Morelos | blue/lower cloister *** | 0.04 | 0.011 | 0.439 | – | – | – | – | – | – | – | – | – | – | – | – | 99.498 |
| Tepoztlan | Morelos | Blue *** | 0.036 | – | 0.686 | 0.013 | 0.015 | – | – | – | – | – | – | – | – | – | – | 99.236 |
| Tepoztlan | Morelos | Stucco | – | – | 0.04 | – | – | – | – | – | – | – | – | – | – | – | – | 99.952 |
| Ocuituco | Morelos | Blue ** | 0.017 | – | 0.078 | – | – | – | – | – | – | – | – | – | – | – | – | 99.897 |
| Ocuituco | Morelos | Stucco | 0.418 | – | 0.024 | – | – | – | – | 0.074 | – | – | – | – | – | – | – | 99.468 |
| Zacualpan de Amilpas | Morelos | Blue~ | 0.046 | 0.016 | 0.876 | 0.024 | 0.015 | – | – | – | – | – | – | – | – | – | – | 99.016 |
| Zacualpan de Amilpas | Morelos | Stucco | 0.047 | 0.01 | 0.805 | 0.022 | 0.014 | – | – | – | – | – | – | – | – | – | – | 99.099 |
| Tepeji del Río | Hidalgo | blue/background * | 3.588 | 0.082 | 0.125 | – | 0.014 | 1.713 | 1.051 | 1.818 | 0.016 | 0.019 | – | 0.082 | 0.014 | – | – | 83.438 |
| Tepeji del Río | Hidalgo | blue/virgin´s mantle * | 2.375 | 0.043 | 0.163 | – | – | 1.8 | 4.47 | 1.112 | 0.027 | 0.021 | – | 0.044 | 0.037 | – | 0.347 | 89.55 |
| Tepeji del Río | Hidalgo | Stucco | 0.017 | – | 0.26 | – | – | – | – | – | – | – | – | – | – | – | – | 99.711 |
| Alfajayucan | Hidalgo | Blue | 0.012 | – | 0.067 | – | – | – | 0.108 | – | – | – | – | – | – | – | 0.036 | 99.766 |
| Actopan | Hidalgo | blue/open chapel *** | 0.03 | – | 0.892 | – | – | – | – | – | – | – | – | – | – | – | – | 99.056 |
| Actopan | Hidalgo | Stucco | 0.031 | – | 0.254 | – | – | – | – | – | – | – | – | – | – | – | – | 99.7 |
| Tezontepec | Hidalgo | dark/blue ** | 0.023 | – | 0.308 | – | – | – | – | – | – | – | – | – | – | – | – | 99.645 |
| Tezontepec | Hidalgo | Stucco | 0.015 | – | 0.079 | – | – | – | – | – | – | – | – | – | – | – | – | 99.645 |
| Tezontepec | Hidalgo | light/blue ** | 0.034 | – | 0.28 | – | – | – | 0.052 | – | – | – | – | – | – | – | – | 99.613 |
| Tepetitlán | Hidalgo | blue/interior | 0.084 | 0.01 | 0.708 | 0.015 | 0.02 | – | – | – | – | – | – | – | – | – | – | 99.155 |
| Tepetitlán | Hidalgo | dark blue/superior part of wall painting | 0.018 | – | 0.189 | – | 0.015 | – | 0.014 | – | – | – | – | – | – | – | – | 99.751 |
| Tepetitlán | Hidalgo | light blue/superior part of wall painting | 0.013 | – | 0.076 | – | 0.013 | – | 0.021 | – | – | – | – | – | – | – | – | 99.868 |
| Tepetitlán | Hidalgo | blue/arch flowers | 0.121 | 0.01 | 0.341 | – | 0.014 | – | – | – | – | – | – | – | – | – | – | 99.499 |
| Actopan | Hidalgo | blue/cloister~ | 0.026 | – | 0.279 | – | – | – | 0.033 | – | – | – | – | – | – | – | – | 99.638 |
| Actopan | Hidalgo | stucco/cloister | 0.031 | – | 0.275 | – | – | – | – | – | – | – | – | – | – | – | – | 99.671 |
| Acatlán | Hidalgo | Blue *** | 0.069 | 0.074 | 0.567 | 0.01 | 0.011 | – | – | – | – | – | – | – | – | – | – | 99.266 |
| Acatlán | Hidalgo | Stucco | 0.033 | 0.05 | 0.28 | – | – | – | – | – | – | – | – | – | – | – | – | 99.28 |
| Tlaltizapan | Morelos | blue/chapel *** | 0.028 | – | 0.235 | – | – | – | – | – | – | – | – | – | – | – | – | 99.723 |
| Tlaltizapan | Morelos | stucco/chapel | 0.018 | – | 0.045 | – | – | – | – | – | – | – | – | – | – | – | – | 99.928 |
| Tlaquiltenango | Morelos | Blue *** | – | – | 0.331 | – | – | – | – | – | – | – | – | – | – | – | – | 99.659 |
| Tlaquiltenango | Morelos | Stucco | – | – | 0.068 | – | – | – | – | – | – | – | – | – | – | – | – | 99.922 |
| Tepeapulco | Hidalgo | Blue *** | 0.031 | 0.014 | 0.333 | – | – | – | – | – | – | – | – | – | – | – | – | 99.608 |
| Tepeapulco | Hidalgo | Stucco | 0.014 | 0.072 | – | – | 0.025 | – | – | – | – | – | – | – | – | – | – | 99.879 |
| Zempoala | Hidalgo | Blue~ | 0.028 | – | 0.27 | – | – | – | – | – | – | – | – | – | – | – | – | 99.682 |
| Zempoala | Hidalgo | Stucco | 0.029 | – | 0.259 | – | – | – | – | – | – | – | – | – | – | – | – | 99.68 |
| Epazoyucan | Hidalgo | light blue ** | 0.029 | – | 0.193 | 0.011 | – | – | – | – | – | – | – | – | – | – | 0.016 | 99.727 |
| Epazoyucan | Hidalgo | dark blue ** | 0.018 | – | 0.256 | – | – | – | 0.092 | – | – | – | – | – | – | – | 0.018 | 99.593 |
| Epazoyucan | Hidalgo | virgin mantle ** | 0.023 | – | 0.245 | – | – | – | – | – | – | – | – | – | – | – | – | 99.692 |
| Epazoyucan | Hidalgo | Stucco | 0.014 | – | 0.063 | – | – | – | – | – | – | – | – | – | – | – | – | 99.908 |
| Oxtoticpac | Estado de México | Blue ** | 0.045 | 0.011 | 0.385 | – | 0.025 | – | – | – | – | – | – | – | – | – | – | 99.518 |
| Oxtoticpac | Estado de México | blue/intervention ** | 0.043 | – | 0.393 | 0.249 | 0.024 | – | 0.012 | – | – | – | 0.028 | – | – | 0.023 | 0.099 | 99.108 |
| Oxtoticpac | Estado de México | Stucco | 0.021 | – | 0.203 | – | 0.021 | – | – | – | – | – | – | – | – | – | – | 99.7345 |
| Cholula | Puebla | Greenish blue * | 0.087 | – | – | – | 0.013 | – | 0.016 | – | – | – | – | 0.033 | 0.01 | – | – | 99.757 |
| Cholula | Puebla | Greenish blue ** | 0.031 | – | 0.245 | – | 0.012 | – | – | – | – | – | – | 0.072 | – | – | – | 99.621 |
| Cholula | Puebla | Stucco | 0.013 | – | 0.054 | – | 0.013 | – | – | – | – | – | – | – | – | – | – | 99.918 |
| Huejotzingo | Puebla | Blue | 0.03 | – | 0.316 | – | 0.012 | – | – | – | 0.01 | – | – | – | – | – | – | 99.616 |
| Huatlatlauca | Puebla | Blue~ | 0.037 | – | 0.283 | – | – | – | – | – | – | – | – | – | – | – | – | 99.646 |
| Huatlatlauca | Puebla | Stucco | 0.025 | – | 0.239 | – | – | – | 0.022 | – | – | – | – | – | – | – | 0.024 | 99.673 |
| Tecali | Puebla | Blue ** | 0.033 | – | 0.42 | – | 0.031 | – | – | – | – | – | – | – | – | – | – | 99.485 |
| Tecali | Puebla | Stucco | 0.041 | – | 0.303 | – | 0.061 | – | – | – | – | – | – | – | – | – | – | 99.56 |
| Cuauhtinchan | Puebla | dark blue/church ** | 0.019 | 0.02 | 0.448 | – | 0.011 | – | – | – | – | – | – | – | – | – | – | 99.492 |
| Cuauhtinchan | Puebla | light blue/church ** | 0.066 | 0.014 | 0.423 | – | 0.012 | – | – | – | – | – | – | – | – | – | – | 99.479 |
| Cuauhtinchan | Puebla | blue/convent * | 0.023 | – | 0.189 | – | – | – | – | – | – | – | – | – | – | – | – | 99.773 |
| Cuauhtinchan | Puebla | Stucco | 0.024 | 0.017 | 0.284 | – | 0.011 | – | – | – | – | – | – | – | – | – | – | 99.658 |
| Tula | Hidalgo | Blue ** | 0.126 | – | 0.303 | – | – | 0.025 | – | – | – | – | – | – | – | – | – | 99.513 |
| Tula | Hidalgo | Stucco | 0.022 | – | 0.25 | – | 0.016 | – | – | – | – | – | – | – | – | – | – | 99.626 |
| Meztitlán | Hidalgo | Blue *** | 0.039 | – | 0.465 | – | 0.016 | 0.256 | 0.02 | – | – | – | – | 0.398 | – | – | – | 98.793 |
| Meztitlán | Hidalgo | stucco | 0.015 | – | 0.062 | – | – | – | – | – | – | – | – | – | – | – | – | 99.911 |

## Appendix A

Comparison between XRD and XRF analysis showing that even when the amount of Fe is high, the blue samples can be made by Maya blue. LE = Light elements.

**Table A2.** XRF data of blue samples cantaining palygorskite or sepiolite compared with XRF data of stucco without painting. Fe values can be compared.

| Sample | Minerals Identified by XRD | Metallic Elements (XRF) | XRF (%) | Values of Fe |
|---|---|---|---|---|
| Blue Ocuituco (M1) | Calcite: $CaCO_3$ <br> Gypsum: $CaSO_4\ 2H_2O$ <br> Sepiolite:$Mg_4(Si_6O_{15})(OH)_2 \cdot 6H_2O$ | Ti | 0.017 | |
| | | Fe | 0.078 | |
| | | LE | 99.897 | Fe values slightly higher |
| Stucco Ocuituco | – | Ti | 0.418 | |
| | | V | 0.074 | |
| | | Fe | 0.024 | |
| | | LE | 99.468 | |
| Light blue Tezontepec (M2) | Calcite: $CaCO_3$ <br> Gypsum: $CaSO_4 \cdot 2H_2O$ <br> Palygorskite: $(MgAl)_2Si_4O_{10}(OH) \cdot 4(H_2O)$ | Ti | 0.034 | |
| | | Fe | 0.28 | |
| | | Pb | 0.052 | |
| | | LE | 99.613 | |
| Dark blue Tezontepec (M5) | Calcite: $CaCO_3$ <br> Gypsum: $CaSO_4 \cdot 2H_2O$ <br> Palygorskite:$(MgAl)_2Si_4O_{10}(OH) \cdot 4(H_2O)$ | Ti | 0.023 | Higher values of Fe |
| | | Fe | 0.308 | |
| | | LE | 99.645 | |
| Stucco Tezontepec | – | Ti | 0.015 | |
| | | Fe | 0.079 | |
| | | LE | 99.645 | |
| Blue upper cloister Oaxtepec (M3) | Calcite: $CaCO_3$ <br> Gypsum,: $CaSO_4 \cdot 2H_2O$ <br> Palygorskite: $(MgAl)_2Si_4O_{10}(OH) \cdot 4(H_2O)$ | Ti | 0.024 | |
| | | Fe | 0.077 | Lower values of Fe |
| | | LE | 99.881 | |
| Blue lower cloister Oaxtepec (M4) | Calcite: $CaCO_3$ <br> Palygorskite: $(MgAl)_2Si_4O_{10}(OH) \cdot 4(H_2O)$ | Ti | 0.04 | |
| | | Mn | 0.011 | |
| | | Fe | 0.439 | Higher values of Fe |
| | | LE | 99.498 | |
| Stucco Oaxtepec | – | Ti | 0.012 | |
| | | Fe | 0.079 | – |
| | | LE | 99.893 | |

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
