# Peer review of "Maya Blue Used in Wall Paintings in Mexican Colonial Convents of the XVI Century"

_coatings, doi:10.3390/coatings11010088_

Round 1

Reviewer 1 Report

The Maya blue pigment is a composite of organic and inorganic constituents, therefore a comprehensive study on the pigment should imply not only diffraction and XRF or SEM/EDS studies, but also infrared and  Raman spectroscopies at least, the latter two to confirm the nature of the organic component. However this paper is focused on the mineralogical caracterization end elemental composition only and none of the investigated samples has been tested as far as the indigo pigment is concerned. Therefore, without this data the paper is largely incomplete. Many typing or conceptual errors occur somewhere.

Line 16:   o1There    ???? (Typing error?)

Line 57:     the chemical formula of palygorskite is wrong : change “Si8O20Al2Mg2(OH)2(H2O)4.4(H2O)” with (Mg,Al)2Si4O10(OH)·4H2O that is the IMA official formula accepted as reported in Appendix B.

Line 117:  “Program1”     1 is an apex

Line 192:  Why only samples with no needle-shaped crystals were used? To avoid preferred orientation in the XRPD pattern? Please explain.

Line 240:   replace “2” with “two”

Line 354:  “Change “basanite” with “bassanite”

Line 363:  Change “17,86” with “17.86”

Line 372: Table 7:  The unit-cell volume is not in cm3 but in Å3! Moreover it is a non-sense reporting this volume with two decimal digits because of the high standard deviations on the unit-cell parameters. Otherwise show the esd on the volume.

Line 384: Change “Prusian” with “Prussian”

Line 454: replace “SiO2” with “SiO2

Line 502: replace “FE” with “Fe”

Appendix B. All the mineral names should be written in English not in Spanish. Moreover the formulae should be written with the appropriate pedices.

Reviewer 2 Report

This is my report as a reviewer for the manuscript entitled " Maya Blue Used in Wall Paintings in Mexican Colonial Convents of the XVI Century" by Mainou and Co-authors.

The study is very interesting and rich in detailed information. The results of the investigations have important archaeological-economic implications.

few suggestions,

pag. 1 line 16 olThere please correct

pag. 3 line56 remove the bracket

Pag. 5 line 157 I don't understand well please can you rephrase it, (In bold we show the colours that can be related to Maya blue.) there are no parts in bold

Pag. 7 table 3: the unit of measurement in the first image, does it apply to all the others?

Pag. 11 table 5 in de columns, please correct

Pag. 13 line 345 measured should be replaced with characterized

Fig. 8, mark the main peak of the palygorskite and sepiolite with a letter

Fig. 8 I would remove 0.70 nm, 1.0 nm ecc (or specify the meaning in the caption)

Fig. 8 (Co)? I would remove (or specify the meaning in the caption)

Pag. 18 line 454 put the 2 in subscript

In the captions of figures 3 and 4 a minimum description of the images should be integrated

Apart from this minor quibble, the paper looks to be in good shape for publication.

if it can be useful I suggest you read the following article

Chiari, G., Giustetto, R., & Ricchiardi, G. (2003). Crystal structure refinements of palygorskite and Maya Blue from molecular modelling and powder synchrotron diffraction. European Journal of Mineralogy15(1), 21-33.

Reviewer 3 Report

The paper “Maya Blue Used in Wall Paintings in Mexican Colonial Convents of the XVI Century”, by Luisa Straulino Mainou et al, present an interesting work in the field of Maya blue pigment. The paper presents valuable results from the past history by using physics modern methods and devices.

The paper is recommended for publication with minor revisions:

  1. In the abstract, at line 16, appear the word “olThere”. Please correct it.
  2. At line 57, in the theoretical formula of Maya blue appear a small point. Please check again the chemical formula.
  3. The image from Figure 2 is original? If not, please insert a reference.
  4. In section 3.2 are presented the SEM measurements, and based on them, are presented the EDS chemical analyses. My question is related to the ageing processes of the materials. All the cracks, and morphological structures of the samples from SEM are actually the results of ageing in time. Also, being exposed to the atmospheric pressure, in the EDS, the quantities of oxygen and carbon are increased from this exposure to the ambient. How do you comment the influence of the ageing process in your samples for both chemical and morphological investigations?

Round 2

Reviewer 1 Report

With the revisions applied now the paper is suitable for publication